



# Spatial and temporal analysis of extreme sea level and skew surge events around the coastline of New Zealand

Scott A. Stephens[1], Robert G. Bell[1], Ivan Haigh[2]

[1]National Institute of Water and Atmospheric Research, PO Box 11 115, Hamilton 3251, New Zealand

[2] Ocean and Earth Science, National Oceanography Centre, University of Southampton, European Way, Southampton, SO14 3ZH, UK

*Correspondence to*: Scott A. Stephens (Scott.Stephens@niwa.co.nz)

**Abstract.** Coastal flooding is a major global hazard, yet few studies have examined the spatial and temporal characteristics of extreme sea level and associated coastal flooding. Here we analyse sea-level records around the coast of New Zealand (NZ)

to quantify extreme sea level and skew-surge frequency and magnitude. We identify the relative magnitude of sea level components contributing to 85 extreme sea level and 135 extreme skew-surge events recorded in NZ since 1900. We then examine the spatial and temporal clustering of these extreme sea-level and skew-surge events and identify typical storm-tracks and weather types associated with the spatial clusters of extreme events. We find that most extreme sea levels were driven by moderate skew-surges combined with high perigean-spring tides. The spring–neap tidal cycle, coupled with a moderate surge

climatology, prevents successive extreme sea-level events from happening within 4–10 days of each other and generally there are at least 10 days between extreme sea-level events. This is similar to findings from the UK (Haigh et al. 2016), despite NZ having smaller tides. Extreme events more commonly impacted the east coast of the North Island of NZ during Blocking weather types, and the South Island and west coast of the North Island during Trough weather types. The seasonal distribution of both extreme sea-level and skew-surge events closely follows the seasonal pattern of mean sea-level anomaly (MSLA)—

MSLA was positive in 92% of all extreme sea-level and in 88% of all extreme skew-surge events. The strong influence of low-amplitude (−0.06 to 0.28 m) MSLA on the timing of extreme events shows that mean sea-level rise (SLR) of similarly small height will drive rapid increases in the frequency of presently rare extreme sea levels. These findings have important implications for flood management, emergency response and the insurance sector, because impacts and losses may be correlated in space and time.




## 1 Introduction

Coastal flooding is a major global hazard with historical events killing 100,000's of people and causing billions of dollars damage to property and infrastructure (e.g., Lagmay et al. 2015; Needham et al. 2015; Haigh et al. 2016). Globally, it has been estimated that up to 310 million people are already exposed to a 1 in 100-year flood from the sea (Jongman et al. 2012; Hinkel

et al. 2014; Muis et al. 2016). In New Zealand this includes 72,000 people—1.5% of the population (Paulik et al. 2019). This will get worse, since without adaptation, it has been estimated that 0.2–4.6% of global population will be flooded annually under 25–123 cm (RCP2.6–RCP8.5) of global mean SLR (Hinkel et al. 2014). Improved understanding of extreme sea level and coastal flooding events is therefore important.

To understand and manage their risk exposure, central government agencies, environmental and emergency managers and the insurance and financial sectors, all require knowledge of the likely frequency and magnitude of extreme sea-level events and their clustering in time and space. Studies have quantified the frequency and magnitude of extreme sea levels at regional (e.g., Bernier & Thompson 2006; Haigh et al. 2010a; Haigh et al. 2010b; Haigh et al. 2014) and global (e.g., Menéndez & Woodworth 2010; Muis et al. 2016) scales, but, other than Haigh et al. (2016), recognition and analysis of spatial and temporal

extreme sea level characteristics and associated coastal flooding is lacking globally. Haigh et al. (2016) provided an analysis of spatial and temporal extreme sea level characteristics for the UK coastline. In this paper we undertake a similar analysis for the New Zealand (NZ) coastline, contributing to a growing understanding of spatial and temporal extreme sea level characteristics worldwide for different tidal and storm contexts.

In the past, relatively few NZ sea-level records have been included in global extreme sea-level studies (e.g., Wahl et al. 2017) whose focus was on global intra-comparisons and trends. Therefore, here we provide a comprehensive assessment of extreme sea level and skew-surge in NZ based on a more extensive tide gauge dataset (with on-average records of 30-years in length). We use sea-level gauge records to quantify the frequency-magnitude distribution of extreme sea level and skew-surges. Skew-surge is the height difference between a sea-level peak and the nearest high tide. We also examine the influence of the mean

sea-level anomaly (MSLA) on the timing of extreme sea-level events. MSLA is derived from the low-frequency non-tidal residual sea-level (Merrifield et al. 2013), which is dominated by seasonal heating and cooling of the sea (Bell & Goring 1998; Boon 2013) and inter-annual climate cycles such as El Niño–Southern Oscillation (Goring & Bell 1999) and is often not considered in extreme sea level analyses because in most places it is small compared with tide and skew-surge—we show that it has an important influence on the timing of extreme sea-level and skew-surge events in NZ. We analyse the spatial and

temporal clustering of extreme sea-level and skew-surge events and identify typical storm-tracks and weather types associated with spatial clusters of extreme events.



## 2. Data and Methods

We analysed sea-level records from 30 locations in NZ (Figure 1). The duration of the records ranged from 6–115 years with a mean and median of 31 and 20 years respectively, with mostly concurrent records since the early 1990s. The sea-level records were quality-analysed to remove any spikes, timing errors, or datum shifts but leaving any data gaps. Tide gauge stations globally are typically located in protected embayments, which limits the direct impact of wind waves relative to open coastlines, but storm surges (and skew-surges) generally are well sampled by tide gauges (e.g., Merrifield et al. 2013). The NZ sea-level records analysed here are from a variety of locations including wave-exposed open-coast, inside port breakwaters, or mounted on wharves inside estuaries, so they have different levels of wave exposure. The sampling frequency varies between and within individual records, from 1-minute up to 1-hour sampling intervals. At some sites far infra-gravity waves of >2 minutes and several decimetres amplitude are observed in the 1-minute data during local or remote storm events (Thiebaut et al. 2013) and tsunami events are occasionally recorded too. To enable consistent analysis between gauges and within long-duration records with varied sampling frequency, we subsampled all sea-level data to 1-hour intervals after first applying a 15-minute running average to minimise the effect of far infragravity and tsunami waves. Thus, we analyse still-water levels with short-period wave effects minimised. Some open-coast sites can experience wave-setup when large waves are present.

The sea level heights are specified relative to local vertical datum (Hannah & Bell 2012). Average relative mean sea levels in New Zealand have exhibited an approximately linear rise over the last century of 1.7 ± 0.1 mm yr−1 (Hannah & Bell 2012). Therefore, before further analysis the data were linearly detrended to a zero mean relative to local vertical datum for each gauge, to remove the effects of historical SLR from the sea-level distribution and create a quasi-stationary timeseries required for extreme-value analysis (Coles 2001). We used a linear trend rather than removing annual mean sea level because we wished to retain inter-annual sea level variability as a long-period component of MSLA in the storm-tide distribution.

The detrended sea-level records were separated into their main component parts. Tidal elevations were predicted using harmonic analysis, following Foreman et al. (2009) and were subtracted from the detrended sea level to obtain the non-tidal residual. The solar annual and semi-annual tides were omitted from the tidal harmonic predictions because most of the seasonal signal is actually driven by non-astronomical effects like seasonal heating and cooling (Bell & Goring 1998; Boon 2013) and we wished to later analyse seasonal effects on the timing of extreme sea-level events. The mean sea-level anomaly (MSLA) was calculated using a 30-day running average of the non-tidal residual (Haigh et al. 2014)—MSLA is a slowly-varying (≥ 1-month) component of both sea level and skew-surge, which includes seasonal sea-level variability. Skew-surge was calculated as the absolute difference between the maximum recorded sea level during each tidal cycle and the predicted maximum astronomical tidal level for that cycle, irrespective of differences in timing between these (Batstone et al. 2013; Williams et al. 2016)—every high tide has an associated skew-surge. Skew-surge is a relevant metric of surge in tidally-dominant locations





like NZ (e.g., Merrifield et al. 2013) because the extreme sea level and resulting flooding exposure (excluding wave

overtopping) usually occurs for a few hours around the high tide.

Like Haigh et al. (2016), we investigate two types of events: (1) extreme sea-level events (relevant to coastal flooding) that

reached or exceeded the 1 in 5-year return level (the sea level equalled or exceeded once, on average, every 5 years) in the

detrended series (i.e. independent of SLR) and (2) extreme skew-surges that reached or exceeded the 1 in 5-year return level.

Some of the skew-surge events coincide with the extreme sea-level events, when the storm surge occurred around the time of

high water of a spring tide; others do not coincide, because the surge occurred near a small high tide or on a neap tide.

To identify the 1 in 5-year return levels the extremes of the distributions were modelled using a generalized Pareto distribution

(GPD) fitted to peaks-over-threshold (POT) data (Coles 2001). POT consisted of the largest 5 high-water maxima per year,

with the maxima being separated by a minimum of 3 days, since separate meteorological systems generally pass over NZ with

4–7 days of each other. A test using a 2-day threshold for both sea level and skew-surge showed that the identification of

extreme events was insensitive to the time threshold. The GPD model was used to determine the 1 in 5-year return level

thresholds for extreme sea level and skew-surge because the model is fitted to independent POT maxima, the sampling of

which is consistent with the identification of unique extreme sea-level and skew-surge events associated with storms. However,

for predicting long return-period levels such as for 100 years, the GPD model is likely to be biased low when analysing short

sea-level records (Figure 1) due to the likelihood of observing few very large maxima over a short record (e.g., Haigh et al.

2010b). Therefore, we also applied the skew-surge joint-probability method (SSJPM) (Batstone et al. 2013) to determine

extreme sea level frequency and magnitude. Joint-probability methods provide more robust low-frequency magnitude

estimates for short-duration records because they overcome the main theoretical limitations of extreme value theory application

to sea levels—splitting the sea level into its deterministic (predictable) tidal and stochastic (e.g. unpredictable, storm-driven)

non-tidal components, and analysing the two components separately before recombining (e.g., Tawn & Vassie 1989; Haigh et

al. 2010b). Sea level return periods can be estimated from relatively short records because all skew-surge events are considered,

not just those that lead to extreme levels. A limitation of the SSJPM and other joint-probability methods is that it assumes tide

and skew-surge are independent, which has been shown true in the UK (Williams et al. 2016) but has not been fully investigated

in NZ, although comparisons with direct maxima methods for > 50-year-long records (Table S7) give similar results for return

periods ≥ 10 years and thus support the validity of the independence assumption. Santamaria-Aguilar and Vafeidis (2018)

found skew-surge to be independent of high-tide height in mixed semi-diurnal tidal regimes located in regions with a narrow

continental shelf, like NZ. However, this may not be the case inside harbours where storm surge magnitude can depend on the

tidal stage (e.g., Bernier & Thompson 2007; Horsburgh & Wilson 2007; Goring et al. 2011). We provide extreme sea-level

estimates from both GPD and SSJPM (Table S7) for application in coastal flood exposure analyses.



To characterise storms in the wider NZ region, we used Kidson (2000)'s classification of synoptic weather regimes associated with clusters of extreme sea-level and/or skew-surge events that impacted at least three tide gauge sites during the event. Kidson (2000) defined three weather 'regimes', characterized by: (i) frequent low-pressure troughs crossing the country, (ii) high-pressure systems to the north with strong zonal flow to the south of the NZ, and (iii) blocking patterns with high-pressure systems more prominent in the south (Figure S1). Rueda et al. (2019) used a method of weather typing to develop statistical predictors for storm surge and wave height in NZ based on statistical relationship with MSL pressure fields. The Kidson (2000) weather regimes are easily associated with the extreme sea-level and skew-surge spatial clusters identified.

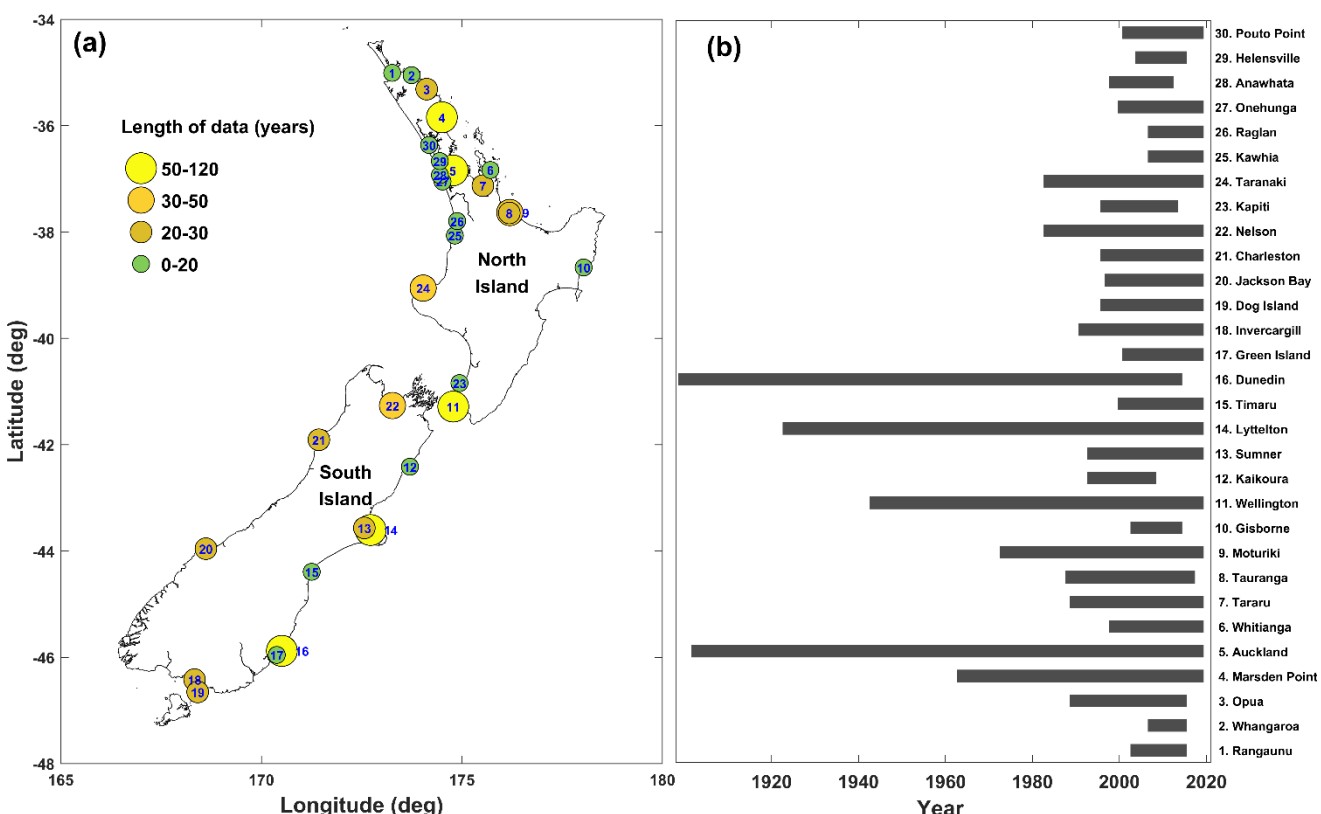

**Figure 1. (a) Location of tide gauge sites around NZ, with site number; and (b) duration of the sea level records.**



## 3. Results

### 3.1 Comparison of events

We compared the dates and return periods for both extreme sea-level and extreme skew-surge events, and ascertained how many skew-surge events also resulted in extreme sea levels. 155 measured sea levels reached or exceeded the 1 in 5-year return level across all 30 study sites (Figure 1). We found that these were generated by 85 distinct storm events (Figure 2a, Table S2). In total, 191 skew-surges reached or exceeded the 1 in 5-year return period across the 30 sites, generated by 135 distinct storm events (Figure 2b, Table S3). This difference in event numbers is consistent with NZ's coastal context of meso- to micro-tidal ranges and moderate storm surges, which are usually several decimetres (not metres) high, meaning tides are a key determinant (or precursor) for coastal flooding rather than solely storm surges.

Like Haigh et al. (2016), there are unavoidable issues with the database that arise because hourly tide gauge records do not cover all the full 118-year period analysed, since the start of the Auckland and Dunedin records in 1900. It is obvious, examining Figure 2c, that we are likely missing many events before the early 1990s, when records were spatially sparse. The decline in data availability post-2010 occurs due to discontinuation of some sea-level gauges around that time, but there are also some recent records that we were unable to obtain for analysis. The highest return period from two notable historical storms is probably lower than it should be; e.g., the great 1936 ex-tropical cyclone (Brenstrum 2000), and ex-tropical cyclone *Gisele* in 1968 (Revell & Gorman 2003). Although we have data at some sites for these events, tide gauges were not necessarily operational at the time along the stretches of the coastline where the sea levels or skew-surges were likely to have been most extreme; this is the case for the 1936 (except in Auckland) and 1968 events. Notwithstanding these issues, our analysis of the events that we have on record does provide important insights, as described below.



**Figure 2. (a) Return period of the highest sea levels in each of the 85 extreme sea-level events, offset for mean sea level; (b) return period of the highest skew-surges in each of the 135 skew-surge events; (c) the number of sites per annum for which sea level data is available across the 30 sites; and (d) pie chart showing the number of skew-surge events that led to extreme sea-level events (blue) and the number that did not (green).**

It is not possible for tide alone to result in a ≥ 1 in 5-year return period extreme sea level, so all observed sea-level events were associated with storms that produced skew-surge. However, only 29 of the 135 (21%) extreme skew-surge events led to extreme sea levels (Figure 2d), while the majority (79%) did not. Hence, as Haigh et al. (2016) found for the UK coast, most extreme sea levels arose from moderate (i.e. <1 in 5-year return levels) skew-surge events combined with astronomical spring high tides. Occasionally a smaller high tide combined with a large skew-surge to produce an extreme sea level—more so for sites with a micro-tidal range (e.g., Wellington, see Figure 3). The ratio of maximum observed skew-surge to maximum observed sea level ranged from 0.23–0.51 with a median of 0.34 (Figure S2). The tide is usually the dominant component of sea-level height, even during extreme events, and extreme sea levels occur close to high tide and are unlikely during a small





165   neap high-tide. Extreme sea level elevation is linearly related to mean high water spring (MHWS) elevation (Figure 4). Equations 1 and 2 are the least-squares linear fits between the MHWS-7 (the height equalled or exceeded by the highest 7% of all high tides) and the 5-year and 100-year return sea level, respectively. Outliers to the linear fits occur in some upper-estuarine locations where the skew-surge/sea-level ratios are relatively large (Figure S2).

5-year return period extreme sea level (m) = 1.20 × MHWS-7 + 0.23     (1)

170   100-year return period extreme sea level (m) = 1.32 × MHWS-7 + 0.28     (2)

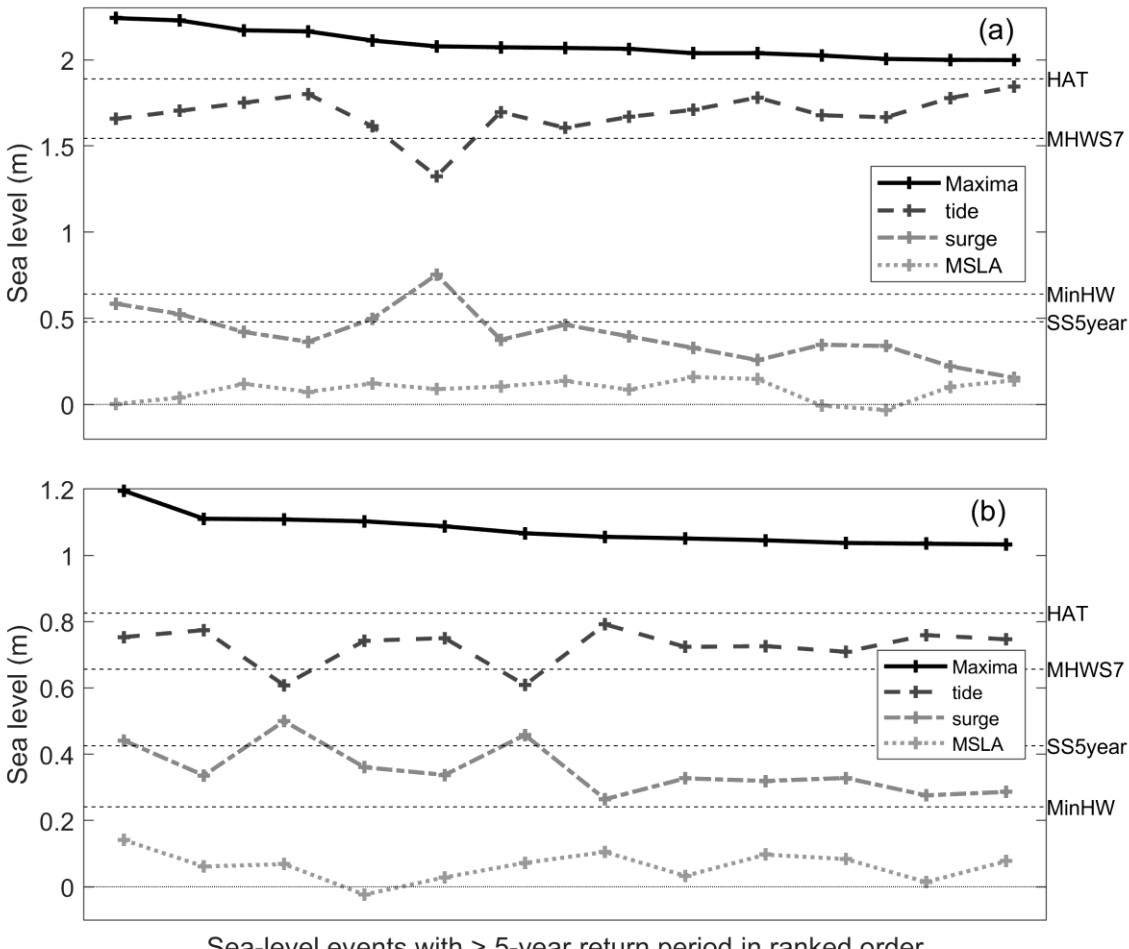

**Figure 3. The contributions of tide, skew-surge and MSLA to the ≥ 1 in 5-year return period extreme sea-level events at two of NZ's longest sea-level recorders with a meso-tidal (a. Auckland) and micro-tidal (b. Wellington) range. HAT = highest astronomical tide, MHWS-7 is the height equalled or exceeded by the highest 7% of all high tides, MinHW = lowest high tide height, SS5year is the**
175   **height of the 1 in 5-year return period skew-surge.**





During the highest extreme sea-level events, all sea-level components tended to be large and positive (Figure S3), with the tide being the largest component, but skew-surge and MSLA being relatively larger in the highest events (Figure S4). The MSLA was positive in 92% of all extreme sea-level and in 88% of all extreme skew-surge events.

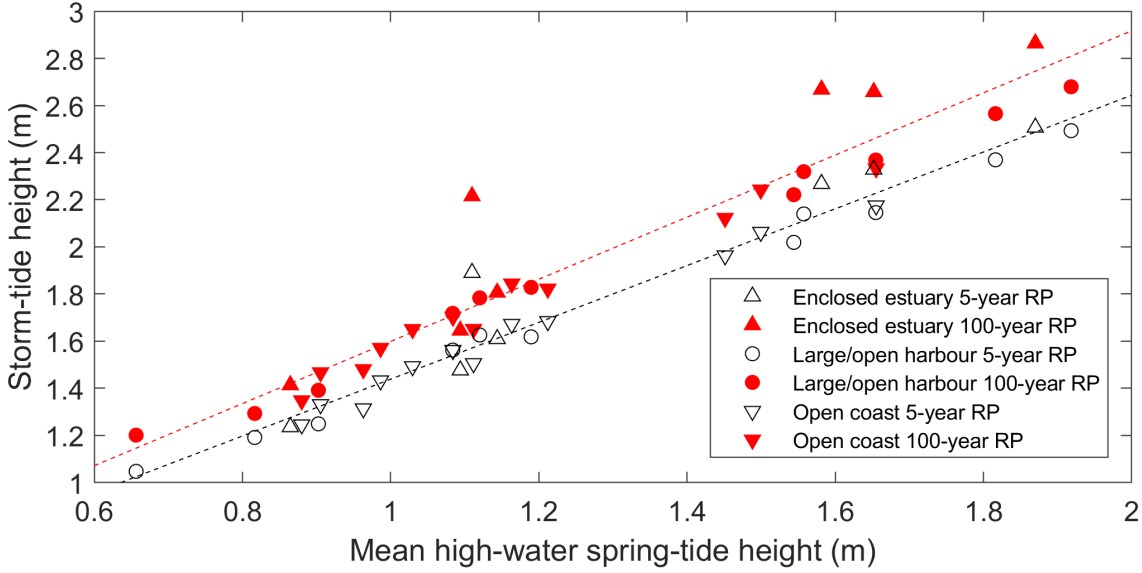

**Figure 4. Linear relationship between high-tide and storm-tide in NZ. MHWS-7 is the height equalled or exceeded by the highest 7% of all high tides. RP = return period, also known as average recurrence interval.**

### 3.2 Spatial analysis

We then considered the spatial characteristics of events around the coast. As expected, and as Haigh et al. (2016) noted for the UK, there is a significant (95% confidence level) correlation (0.50 for the sea-level events and 0.47 for the skew-surge events) between the highest return period of the events and the number of tide gauge sites impacted (Figure S5).

To investigate the different spatial extents of coastline affected, we identified 30 storm events that impacted three or more gauge sites concurrently (Figure 5). We observed two separate categories of spatial footprints, which can be related to the Kidson (2000) weather regimes (Figure S1) as follows:

1. Category 1 footprints predominantly impacted the northeast coast of the North Island and occasionally the west coast of the North Island (see Figure 1(a)). Category 1 footprints were most commonly associated with Blocking weather types (Figure 5), low-pressure systems that tracked from north of NZ (Figure 6) and intensified next to a blocking high-pressure system lying east of NZ. Category 1 footprints had storm centres that lay over the North Island or north of NZ at the time of maximum sea level or skew-surge, with a mean position located just off the northeast coast of the North Island (Figure 6).

2. Category 2 footprints impacted everywhere but mainly the South Island (Figure 1a) and the west coast of the North Island (Figure 5). Category 2 footprints were generally associated with Trough weather regimes, which were the most





common event drivers. Troughs were associated with storm centres that tracked eastwards across NZ, usually across or south of the South Island and always south of the northernmost tip of NZ (Figure 6). Category 2 footprints had

storm centres at the time of maximum sea level that lay in an arc across the South Island from northwest to southeast, with a mean position located south-east of the South Island (Figure 6).

The largest events with return periods ≥ 1 in 50 years were more common on the east coast of the North Island during Blocking weather types, whereas Trough weather types were more common in the South Island and along the west coast of the North Island (Figure 5).




**Figure 5. The spatial footprints of all the sea-level or skew-surge events that impacted at least three tide gauge sites. Red circle = extreme sea level; blue circle = extreme skew-surge. Circle diameter matched to return periods of 5–20, 20–50 and ≥ 50 years (small to large respectively). The spatial cluster category (1 or 2), Kidson weather type and date during each event are shown.**





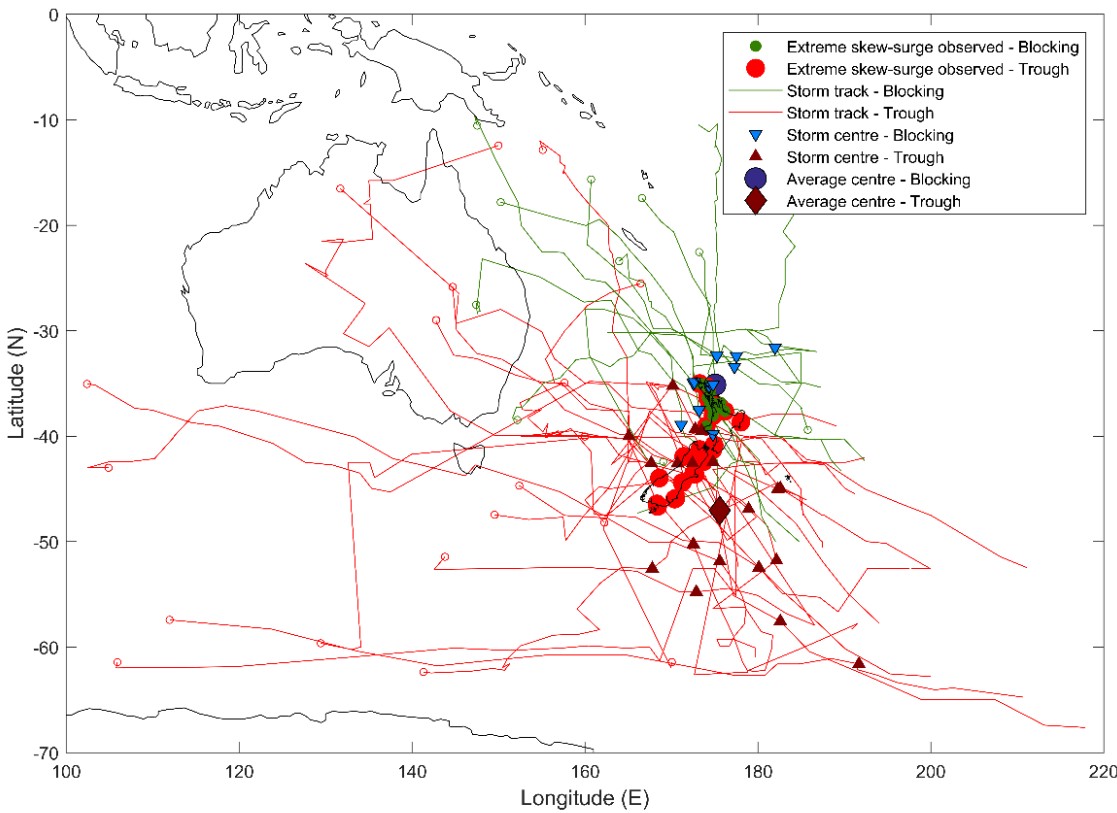

**Figure 6. Storm tracks and weather types contributing to all the sea-level or skew-surge events that impacted at least three tide gauge sites. Storm tracks and the location of the storm centre at sea-level event peak are marked for both Blocking (green/blue) and Trough (red/brown) Kidson (2000) weather types. Open circles denote storm track origin.**

## 3.3 Temporal analysis

Next, we examined the temporal variation in events. Extreme sea-level events with ≥ the 1 in 5-year return level occur throughout the year but are least frequent in late winter and early spring (September–December) (Figure 7). Extreme sea-level events with ≥ 1 in 25-year return level exhibit a peak in January that tapers off through to May followed by another peak in June that tapers off through to September. No sea-level events with ≥ 1 in 25-year return level were observed in October–December. Like extreme sea level, skew-surges with ≥ 1 in 5-year return level are also least likely in the latter half of the year and were most frequent March–July.





The largest tidal constituents around NZ are the M2, S2, and N2 semi-diurnal constituents. The seasonal pattern in the height of the highest tides, which peak around March and September (Figure 7d), is affected by the solar equinox. However, the

seasonal pattern in the number of extreme sea-level and skew-surge events does not follow the highest tides but appears to follow the seasonal pattern of MSLA; MSLA being a component of both extreme sea-level and skew-surge elevations. The mean annual MSLA cycle is dominated by thermo-steric sea-level adjustments and secondary-forcing variables of barometric pressure and alongshore wind stress (Bell & Goring 1998), and it peaks between March and June in NZ—at 22 of 30 sites it peaks in May due to the thermosteric lag. The range of the 95% confidence intervals of the annual sea-level cycle was about

−0.15 to +0.15 m around NZ, and the median amplitude across all sites was 0.04 m (Figure S6). Yet MSLA can be larger in any single month and tended to be larger during observed extreme events—MSLA during events ranged from −0.06 to 0.28 m (Table S2 and Table S3). The MSLA influence on the timing of extreme sea level is perhaps surprising given the dominance of tide on the sea-level elevation (Figure 3 & Figure 4). But because two spring-tidal cycles per month provide sufficient monthly "exposure potential" for sea-level events, the seasonal pattern is then strongly influenced by the mean annual MSLA

cycle. Once the effect of the mean annual MSLA cycle is removed, the seasonal distributions of both extreme sea-level and skew-surge events are more uniform throughout the year (Figure S7 and Figure S8), although there is still a tendency for a lower number of extreme sea-level and skew-surge events near the end of the year in October–December.

In terms of inter-annual climate effects on extreme sea levels, we found no relationship between extreme sea-level height or

return period and large-scale climate modes, such as the El Niño-Southern Oscillation (ENSO) and Southern Annular Mode (SAM), (e.g., Figure S9, Table S10 (Trenberth, 1976)). This aligns with Lorrey et al. (2014)'s findings that ex-tropical cyclone tracks and locations relative to Auckland at their point of closest passage are not directly linked to the phase of ENSO or SAM, but are linked more closely to atmospheric circulation conditions that develop in the Southwest Pacific, to the north of, and over New Zealand. In other words, extreme sea-level events are influenced by random weather events rather than large-scale

climate variability modes.





**Figure 7. Seasonal distribution of sea-level components associated with extreme sea-level and skew-surge events. (a) extreme sea level (b) extreme skew-surge (c) MSLA (d) high-tide ≥ 93rd percentile (includes all high tides, not just those associated with extreme sea-level events). N = number of occasions.**

Following (Haigh et al. 2016), we examined temporal clustering by considering the number of days between consecutive

events. There are no instances of sea-level events happening within 4–10 days of each other, whereas there are occasions when





pairs of skew-surge events occurred within that time interval (Figure 8). Only once did two extreme sea-level events occur within < 4-days of each other, when two separate low-pressure systems intensified against a blocking high-pressure system and impacted NZ on 27 and 30 July 2008.

Figure 8. Time between successive events. a) sea level b) skew-surge.




## 4. Discussion

### 4.1 Tidal influence

In NZ, we find that most of the extreme sea levels are driven by moderate skew-surges combined with high spring or perigean-
spring tides, similar to the UK (Haigh et al. 2016). This is despite the fact that the UK has larger tides, and is also the case for
annual maxima around much of the world (Merrifield et al. 2013). We found that extreme sea level elevations are strongly
correlated with MHWS elevation. This affirms current practice in NZ of forecasting "red-alert" tide dates when high-tide peaks
are predicted to be unusually high (Bell 2010; Stephens et al. 2014) (see NZ Storm-Tide Red-Alert Days; National Institute of
Water and Atmospheric Research; www.niwa.co.nz). The red-alert tide concept works in New Zealand because the semi-
diurnal tides dominate sea-level variability and storm surges are limited to mostly < 0.5 m, which is approximately 25% of the
average tidal range (Stephens et al. 2014). Coastal or hazard managers are advised to keep a close watch on the weather for
lower barometric pressure and adverse winds during the red-alert tide days, as even a minor storm or swell event could lead to
inundation of low-lying areas, especially if accompanied by waves (Bell 2010). For example, the highest storm-tide in
Auckland (NZ) on record since 1900 occurred on 23 January 2011, closing major city highways, and causing millions of
dollars of flood-related damage. At 0.4 m the storm surge was moderate but coincided with a predicted "red alert" (high
perigean-spring) tide, and a background MSLA of nearly 0.1 m (25% of the storm-surge) (Stephens et al. 2014), noting that
ongoing SLR increasingly adds to the impact or frequency of flooding (Stephens et al. 2018).

The observed linear relationships between extreme sea level and MHWS elevation have proven useful for estimation of
extreme sea level at locations without long-term sea-level gauges—extreme sea level can be predicted using short-term sea-
level recorder deployments to establish tidal elevations, or a tidal model (e.g., see New Zealand Tide Forecaster;
www.niwa.co.nz). However, relatively large extreme sea level in some upper-estuarine locations plotted as outliers to the
linear relationships (Figure 4, Figure S2). Research has shown that non-linear interactions between tide, wind and morphology
influence surge generation inside enclosed estuaries (e.g., Plüß et al. 2001; Rego & Li 2010; Orton et al. 2012), a subject for
further research in NZ since there are few long-term sea-level records to help quantify extreme sea level frequency and
magnitude in NZ upper-estuarine locations.

### 4.2 MSLA influence

The median amplitude of the mean annual sea-level cycle from all sea-level records is only 0.04 m, yet the annual MSLA cycle
is a strong influence on the monthly timing of extreme sea-level and skew-surge events. MSLA was positive in 92% of all sea
level and in 88% of all extreme skew-surge events and is clearly an influential component of the highest sea levels (Figures
S3–S4, S7–S8). The sensitivity of extreme sea level to relatively small MSLA shows that small increases in mean sea level,
from long-term SLR, will cause a large increase in the frequency of extreme sea level, a result consistent with other studies
(e.g., Hunter 2012; Sweet & Park 2014; Stephens et al. 2018). Stephens et al. (2018) showed that tidally-dominated sites like



NZ, where the tide is the key determinant of extreme sea level, will be more sensitive to SLR in terms of the increasing
frequency of extreme sea-level events than surge-dominated sites. This is because tidally-dominated sites have a relatively flat
upper tail on the extreme sea level frequency-magnitude distribution. Many parts of the world are even more tidally-dominant
than NZ (Merrifield et al. 2013; Rueda et al. 2017) and will experience earlier emergence of flooding due to greater sensitivity
to SLR (Hunter 2012).

**4.3 Surge influence**

NZ is similar to the UK in that temporal clustering of sea-level events is controlled by the spring-tide cycle (Haigh et al. 2016),
with sea-level events generally not occurring within 10 days of each other historically. However, it only requires storm surge
to elevate high tides to an extreme sea level. Therefore, extreme sea-level and skew-surge events do not strike NZ-wide but
exhibit spatial clustering associated with storm tracks that have general patterns related to weather types.

During the spring and summer of 2017 and 2018 several large storms including ex-tropical cyclones Fehi, Gita, and Hola
struck NZ, most of them coinciding with high perigean-spring tides, causing flooding to homes and damaging infrastructure.
Other notable historical coastal flooding events in NZ occurred in January 2011, during cyclone Gisele in 1968 (de Lange &
Gibb 2000), May 1938 in the Hauraki Plains (Stephens 2018) and during the great cyclone of 1936 (Brenstrum 2000), but the
spatial effects of these historical storms are not well recorded since many sea-level gauges were not operating at those times
(Figure 1). The calculated extreme sea-level and skew-surge frequency and magnitudes (Table S7 and Table S8) are based on
available digital records but could be biased low in places where historical storms are not included. A notable example is in
Tauranga Harbour where the highest skew-surge ever recorded in NZ by a sea-level gauge occurred during cyclone Gisele in
1968 and reached 0.88 m above the predicted high tide (de Lange & Gibb 2000). However, that tide gauge is no longer
operating and hourly data was not available for inclusion in this analysis. We applied a joint-probability method to determine
frequency-magnitude of extreme sea level (Table S7), which partially overcomes the limitation of short-duration sea-level
records.

Before the analysis presented here, the largest recorded skew-surge was 0.88 m measured in Tauranga Harbour in 1968. We
observed 15 larger skew-surges at enclosed estuarine sites (Figure 1a): 1 at Helensville (site 29), 1 at Raglan (site 26), 2 at
Kawhia (site 25) and 11 at Invercargill (site 18). The maximum skew-surge from our analysis was 1.15 m at Raglan on 06
May 2013 and a skew-surge of 1.26 m was inferred from anecdotal evidence in the northern Hauraki Plains on 4 May 1938
(Table S2, Stephens (2018)). However, a much larger non-tidal residual sea level (NTR) was measured at the open-coast
Jackson Bay site around low-tide, during ex-tropical cyclone Fehi on 1 February 2018 (Figure S11). NTR is the difference
between the measured sea level and the predicted tide and is calculated over the full tidal cycle at the sampling frequency of
the sea-level record, hourly in this case. The NTR at Jackson Bay on 1 February 2018 appeared to be a meteorologically-
forced wave that peaked at 2.29 m near low-tide (Figure S11) and would have created a much higher sea level if it had peaked
at high tide—the skew-surge was 0.69 m. Although this was the highest sea-level on record at Jackson Bay, it was a near miss





to a much higher sea level. We found that the "potential" sea level, constructed from the sum of the maximum observed tide and skew-surge, was 7–20% higher than the estimated 100-year return period extreme sea level (Table S9). Hence the potential exists for higher and more damaging sea levels than historically observed in the patchy NZ records, even without considering

the effects of future SLR. Alongside the provided frequency-magnitude estimates, it would be sensible to consider these "potential" extreme sea-level elevations when planning or managing low-lying coastal land use.

We analysed still-water levels with wave effects removed from the record (apart from wave setup that may be implicit in measurements for some events at open-coast gauges e.g., site 8). Waves are often present during storms and wave setup and

runup can raise the water level at the coast substantially, especially on steeper beach gradients or steep-face structures such as rock revetments or seawalls (e.g., Stockdon et al. 2006; Stephens et al. 2011). Hazard analyses should account for wave effects, including far infragravity surges, in wave-exposed locations, on top of the extreme sea level presented here. An analysis of the spatial and temporal clustering of extreme wave events could also be undertaken, following the approach of Santos et al. (2017).


We analysed extreme sea level and skew-surge for 30 sea-level records of varying length throughout NZ. Numerical models, properly calibrated, provide a means of extending the extreme sea-level analyses to develop probabilistic assessments of total water level along the entire coastline. These can include combinations of tidal (e.g., Walters et al. 2001), storm-surge and wave models (e.g., Gorman et al. 2003; Rueda et al. 2019), which can include both hindcasts and climate-change future-casts

(Cagigal et al. submitted). However, numerical storm-surge models for climate change typically do not include MSLA, which is an important component of extreme sea level (for tidally-dominant coasts) and hence flooding exposure (e.g., Stephens et al. 2014; Sweet & Park 2014), and which we have shown to be an important influence on the timing of extreme events. Efforts to forecast MSLA are being made (Widlansky et al. 2017) but work is required to include MSLA into models for the development of probabilistic assessment of extreme water levels to assess coastal hazard risks, taking into account SLR.

**5. Conclusions**

In this paper we analysed sea-level records to quantify extreme sea-level and skew-surge frequency and magnitude around the coast of NZ for the first time. We identified the relative magnitude of sea-level components contributing to 85 extreme sea sea-level and 135 extreme skew-surge events ($\geq$ 5-year return period), which have been recorded in NZ since 1900. We examined spatial and temporal clustering of extreme sea-level and skew-surge events and identified typical storm-tracks and

weather types in the SW Pacific associated with spatial clusters of extreme events.

We found that around NZ most extreme sea levels are driven by moderate skew-surges combined with high perigean-spring tides. Extreme sea-level elevations are highly correlated with and linearly related to mean spring high-tide elevation, enabling



prediction of extreme sea-level elevations in ungauged areas from tidal information, which is readily available from short-term
measurements or tidal models.

The spring-neap tidal cycle, coupled with a small to moderate skew-surge climatology, prevents successive extreme sea-level
events from happening within 4–10 days of each other. Of 85 extreme sea-level events, only once did two extreme sea levels
occur less than 4-days apart due to two closely-spaced storms straddling a peak spring tide period. Generally, there are at least
10 days between extreme sea-level events.

Extreme events caused by "blocking" weather-types coincided with storm centres located in the north of NZ and predominantly
impacted the northeast coast and occasionally the west coast of the North Island. Extreme events caused by "trough" weather-
types coincided with storm centres that track further south, across or south of NZ and impacted mainly the South Island and
the west coast of the North Island.

Lower numbers of extreme sea-level and skew-surge events were observed in late spring to early summer (which is also outside
the ex-tropical cyclone season). This reflects a tendency for more extreme surges earlier in the year but was also noticeably
influenced by the mean annual sea-level cycle that peaks in the austral autumn in most places. There is no apparent relationship
between extreme sea-level events and large-scale climate-variability modes such as ENSO. The considerable influence of the
relatively low-amplitude mean sea-level anomaly shows that a relatively small SLR will drive rapid increases in the frequency
of presently rare extreme sea levels.

**Data sets**

Some of the sea-level records used for the work are available from the Global Sea Level Observing System website run by the
University of Hawaii Sea Level Center. Other records are privately owned and are available from NZ port companies by direct
request. Extreme event data (for reproduction of plots within the manuscript) are included in the Supplementary Tables.

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

**Author contribution**

SS designed the concept with input from RB and IH. IH provided Matlab code. SS undertook the analyses and wrote the paper, with input from RB and IH.

**Competing interests**

The authors declare that they have no conflict of interest.

**Acknowledgments**

SS and RGB were funded by the New Zealand Ministry of Business, Innovation and Employment under Strategic Science Investment Fund (Projects CAVA1904 and CARH2002—National Institute of Water and Atmospheric Research). Benjamin Robinson processed sea-level data. Sea-level data were obtained from various port companies in New Zealand.