# Peer review of "Spatial and temporal analysis of extreme storm-tide and skew-surge events around the coastline of New Zealand"

_Natural Hazards and Earth System Sciences, 2019_

## Referee Comment (RC1) · Anonymous Referee #1 · 23 Dec 2019

**GENERAL COMMENTS**

In this paper, the authors analyse sea level and skew surge extremes (values greater than the 5-yr return level are considered extremes) from 30 tide gauges around the coast of New Zealand.

The objective is to characterise the frequency and magnitude of these extreme events and also estimate the contribution of each sea level component (tide, surge and MSLA) to the sea level extremes. Sea level rise is not taken into account in this study.

In my opinion, the manuscript is very well written, well structured, clear and easy to understand. However, I have one main scientific concern and a few smaller issues that

are explained below.

SPECIFIC COMMENTS

I have one main concern regarding the extremes' methodology: The authors say they use a GPD+POT model to obtain the return levels, but they define POT as the 5 largest events per year (line 99), which in my opinion is the r-largest method and not POT. The POT method should keep a constant threshold through time and in this case it varies each year. I think that extremes selected with the r-largest method should be fitted to a GEV instead to a GPD, so I am not sure if the return levels obtained here are correct.

lines 115-116: It is difficult to extract this information from table S7, maybe those TGs longer than 50 yr could be highlighted or the table could be order by the TGs length instead of the site number?

Table S7: It is not clear to me what the "model percentile" means in table S7. Are 2.5% and the 97.5% the confidence intervals? maybe the table could be simplified using the GPD or SSJPM values +- the confidence intervals?. Why is it that this information does not appear in table S8 for the skew surge?. Another main concern I have is that the return level estimations should not exceed 4 times the length of the observations (Pugh & Woodworth, 2014), since the longest TG is 120 yr, the 1000 return level is not reliable for any location. Maybe the reliable return levels, at each location, could be highlighted. Also, if sea level rise is not included, maybe there is no point in obtaining such large return periods.

lines 134-135: I am not sure if I understand: Some of the 155 SL extremes are not independent, so after keeping only those separated more than 3 days, it results in 85 independent events, is that right? (same for the Skew surges?). Also, which methodology (GPD or SSJM) is used for obtaining the return periods in table S2 and S3?.

line 163: from figure S2 I am not able to infer those values. For example, the ratio for station 18 should be aprox. 1.2m/2m, right?, but figure S2c reads ratio equal to 1.

lines 169-170: Are those equations used somewhere else? what info can we infer? maybe they are not needed?

figures S3b and S4a shouldn't be identical?

figures 7a and S7a shouldn't be identical?

lines 252-254: I am not able to extract this information from figure 8a

line 269: maybe this event could be highlighted in figure 3a

TECHNICAL CORRECTIONS

Figure 4: x-axis is MHWS or MHWS-7?. y-axis is extreme sea level? I think storm-tide is not defined in the paper.

line 219-220: September-December?

line 221: April-July?

line 228: April-July?

---

## Referee Comment (RC2) · Franck Mazas (Referee) · 26 Dec 2019

Overview

The present paper addresses the spatial and temporal patterns of extreme sea levels and skew surges in New Zealand, which is interesting, not so much for providing local extreme values but rather for understanding the conditions in which such events occur. Accounting for MSLA in particular provides valuable information, and finding that the seasonal patterns of extreme sea levels follow the seasonal pattern of MSLA rather than that of astronomical tide is very interesting.

[Figure]

The paper is well written and structured, and the authors are well known in the field of research. Although it could be directly accepted, I have identified several ways of improving it.

Specific comments

First, the reader would probably welcome, after the introduction section, a short description of tides and surge climate in New Zealand. Is the tide semi-diurnal, with diurnal inequality, mixed...? What is the typical tidal range, let us say along both the east and west coast? What are the typical surge values? Is there a wide or a narrow continental shelf? Some of these information are provided here and there in the text, but having a good (though concise) overview before beginning the data / methodology section would help.

As regards the extrapolation technique, it looks like the sampling method is the R-largest method, not the POT one. So in this case there is no Poisson assumption, and this is generally a GEV distribution that is fitted. But maybe is it 5 values per year in average? However, I am more bothered by the choice of considering equally a direct and in indirect approach for extreme sea levels (even more when one of the co-authors has shown the differences between both approaches!). I do not really see what the results of the direct POT (or R-largest) extrapolations provide to the paper, apart from confusion.

I have seen at several occurrences the (widely spread) confusion between level, a vertical position that is always referenced to a datum such as LVD, CD or MSL, and height, a vertical distance, in m, independent of the datum (a height being a difference between two levels). In particular, comparisons between a surge (which is an extra height) and a sea or tidal level is, strictly speaking, physically meaningless. Therefore, figures and tables referring to "sea-level (m)" cannot be understood. For the purpose of this paper, it seems that the relevant physical quantities are the tidal amplitude, or semi-range (difference between tidal level and MSL), skew surge (unchanged), MSLA

(unchanged) and what could be referred to as "sea-level height" (or, preferably, a better name), namely the difference between sea level and MSL. Therefore, "sea-level height" would be the sum of tidal amplitude and skew surge (which includes MSLA), and all quantities could be compared. This could be explained in the first sections, before the results.

Last, it is explained that "red-alert" forecasts are emitted to coastal and hazards managers, because extreme sea levels tend to occur in conjunction with high tides, as shown in this study. But considering the other findings of the study and the relationships with the weather types, the reader would think that it should be rather easy and straightforward to improve this warning system by combining the occurrence of high tides with the weather types prone to surges. This would be an immediate benefit of the study, and it seems strange that this possibility is not mentioned.

- l. 62: ". . . are well sampled by tide gauges": if no seiching occurs, or, if it does occur, if the sampling rate is fine enough to characterize these LF waves

- l. 79/80: how many components for the harmonic analysis?

- l. 81: "annual and semi-annnual tides" -> ". . . components"?

- l. 82: ". . .most of the seasonal signal is actually driven by non-astronomical effects": well, at least not directly, I believe this is a longstanding debate. . . But this is not very important.

- l. 93: this is not the best definition of return period, it would be better to speak of a 1/5 annual probability of exceedance, while highlighting the cumulative effect, year after year, and the probability of encounter in a given period of several years.

- l. 113-116: simple statistics exist for assessing the tide-surge dependence, see in particular Dixon and Tawn (1994)

- Figure 2, legend: if I understand, well, these are not the "85 extreme sea-level events" or "135 skew-surge events", but rather the 85 (135) storms generating the extreme sea

levels (skew surges)

- l. 162/163: "ratio of maximum observed skew surge to maximum observed sea level": only relevant if considering the elevation relative to MSL, see comment above

- Figure 3: the y-axis cannot be "sea level (m)", but something like "surge / water height above MSL", see comment above

- l. 169/170: same comment, is it a comparison of levels in m LVD, or of water height above MSL?

- l. 260: "despite the fact that the UK has larger tides": yes, but laybe the ratio surge / tide is similar?

- l. 264/266: interesting information, but it is mentioned too late in the paper (see comment about a description of tide and surge climate in NZ)

- Section 4.3, §2 and 3: the logical link between the two paragraphs is not clear to me

- l. 325/326 (hourly surges), in relation with l. 336/338 (numerical models): indeed, if the total signal of surge can be accurately distinguished (sometimes very difficult with gauge measurements, hence the skew surge approach, but straightforward through numerical modelling), then much more information can be used for probabilistic extrapolation. Mazas et al. (2014) provide a POT-JPM model that works for the total signal of skew surge and hourly (or 10 min) residual as well.

---

## Author Comment (AC1) · 28 Jan 2020

Also see attachment

Anonymous Referee #1 GENERAL COMMENTS In this paper, the authors analyse sea level and skew surge extremes (values greater than the 5-yr return level are considered extremes) from 30 tide gauges around the coast of New Zealand. The objective is to characterise the frequency and magnitude of these extreme events andalsoestimatethecontributionofeachsealevelcomponent(tide,surgeandMSLA) to the sea level extremes. Sea level rise is not taken into account in this study. In my opinion, the manuscript is very well written, well structured, clear and easy to understand. However, I have one main scientiïñĄc concern and a few smaller issues that are explained below. SPECIFIC COMMENTS I have one main concern regarding the extremes' methodology: The authors say they use a GPD+POT model to obtain the return levels, but they deïñĄne POT as the 5 largest events per year (line 99), which in my opinion is the r-largest method and not POT. The POT method should keep a constant threshold through time and in this case it varies each year. I think that extremes selected with the r-largest method should be ïñĄtted to a GEV instead to a GPD, so I am not sure if the return levels obtained here are correct. Response: We did use the POT method with a fixed height threshold for each site but had not accurately explained it in the text, which made it appear as if we used an r-largest method when in fact we used POT. We have amended the text as shown below, to accurately describe what we did. For further confirmation, in Table 1 below we have also included the number of maxima exceeding the (1.69 m) threshold at Auckland, wherein the number of exceedances of the threshold is seen to vary year to year.

lines 115-116: It is difïñĄcult to extract this information from table S7, maybe those TGs longer than 50 yr could be highlighted or the table could be order by the TGs length instead of the site number? Response: Rather than refer to Table S7, we have added a new supplementary Figure S1 (other supplementary figures sequentially renumbered) which demonstrates how the SSJPM and GPD models are similar for extreme sea levels ≥ 5-year return period but the SSJPM generally better matches the larger out-lying maxima in the longer records. Table S7: It is not clear to me what the "model percentile" means in table S7. Are 2.5% and the 97.5% the conïñĄdence intervals? maybe the table could be simpliïñĄed using the GPD or SSJPM values +- the conïñĄ-dence intervals?. Response: We have added a description to the caption to make it clear that 50% = median of the fitted distribution, 2.5% = lower 95th percent confidence interval, 97.5% = upper 95th percent confidence interval. Simplifying the table using a $\pm$ confidence interval is not possible because the confidence limits are not symmetrical about the median. We have simplified the Table to remove the GPD results, following

review comments from Franck Mazas. Why is it that this information does not appear in table S8 for the skew surge?. Response: the SSJPM method cannot be applied to skew-surge, which is a component of extreme sea level. But we have now included the 95% confidence intervals for the skew-surge in Table S8. Thanks. Another main concern I have is that the return level estimations should not exceed 4 times the length of the observations (Pugh & Woodworth, 2014), since the longest TG is 120 yr, the 1000 return level is not reliable for any location. Maybe the reliable return levels, at each location, could be highlighted. Also, if sea level rise is not included, maybe there is no point in obtaining such large return periods. Response: This is a good point and so we have provided some guidance to temper the use of the return level esti- mates. Using GPD the reliable return levels are about 4 times the record length, but this can be about 10 times the record length for joint-probability methods (see Haigh et al. 2010). Therefore, immediately preceding Table S7, we have added a paragraph to guide the use of Table S7, which says: "Table S7 presents extreme storm-tide return period height estimates out to 1000-year return period. Whereas direct GPD estimates are generally only reliable for return periods up to 4 times the record length (Haigh et al. 2010; Pugh & Woodworth, 2014)†, joint-probability (e.g., SSJPM) estimates can be statistically reliable to about 10 times the record length (Haigh et al. 2010). The record lengths are included in the table to guide the use of extreme storm-tide return period height estimates. For example, the 6-year record length at Whangaroa up to 50-year return period using the SSJPM." We note that return periods are surrogate for probability of occurrence, e.g., 1000-year return period is equivalent to 0.001 annual exceedance probability. Return periods (aka probabilities of occurrence) should not be confused with time periods over which sea-level rise might act. There are several papers showing that show (even quite small) sea-level rise will dramatically change the probability of exceedance of sea-levels reaching a fixed height relative to a land-based datum, but it is still relevant to provide low exceedance probability sea-level height estimates at present-day mean sea level—quantifying how these exceedance rates could change (e.g., Hunter 2010, Sweet & Park 2014, Stephens et al. 2018) is an extra

step beyond the present research paper. lines 134-135: I am not sure if I understand: Some of the 155 SL extremes are not independent, so after keeping only those separated more than 3 days, it results in 85 independent events, is that right? (same for the Skew surges?). Also, which methodology (GPD or SSJM) is used for obtaining the return periods in table S2 and S3?. Response: We have rearranged the text to clarify, as below:

Response: The captions of Tables S2 and S3 now describe which method was used. Thanks. line 163: from fi̧gure S2 I am not able to infer those values. For example, the ratio for station 18 should be aprox. 1.2m/2m, right?, but fi̧gure S2c reads ratio equal to 1. Response: In plot c we had mistakenly plotted maximum skew-surge / MHWS-7. This has now been corrected. Thanks. lines 169-170: Are those equations used somewhere else? what info can we infer? maybe they are not needed? Response: The equations are not used elsewhere in the paper but might be of relevance to a user conducting local studies within NZ, so we prefer to leave them as is. fi̧gures S3b and S4a shouldn't be identical? Response: Figure S3b and S4a (S4b and S5a in revised supplementary information) shouldn't be identical and are not identical. S3b shows high tide at Auckland and S4a shows mean storm-tide elevation throughout NZ. fi̧gures 7a and S7a shouldn't be identical? Response: We deliberately reproduced Figure 7a within Figure S7a for comparison with Figure 7b, to enable the reader to make a quick visualisation of the impact of MSLA on the seasonal distribution of extremes, without having to flick back to the paper . lines 252-254: I am not able to extract this information from fi̧gure 8a Response: The information can be inferred from Figure 8a—there are no events (blue dots) falling within 4–10 days on the y-axis? Haigh et al. 2016 conveyed similar information using the same type of plot (Figure 6, Haigh et al. 2016). line 269: maybe this event could be highlighted in fi̧gure 3a Response: We have altered the text as per below to point to the event in Figure 3a. We have also updated the numbers relative to those originally transferred from Stephens et al. (2014) to report skew-surge instead of storm-surge and have updated the value for MSLA (Stephens et al. 2014 used a wavelet filter rather than a 30-day running average).

[Figure]

TECHNICAL CORRECTIONS Figure 4: x-axis is MHWS or MHWS-7?. y-axis is extreme sea level? I think storm-tide is not defined in the paper. Response: We have re-labelled the axes. Storm-tide is now defined in the first paragraph of the Data and Methods section. line 219-220: September-December? Response: Correct, thanks—text amended to September–December. line 221: April-July? Response: March–July are consecutive months that have higher monthly occurrences of ≥5-year ARI skew-surges than any other month line 228: April-July? Response: March–June is correct based on the peak of the mean annual cycle (not shown).

Please also note the supplement to this comment:
https://www.nat-hazards-earth-syst-sci-discuss.net/nhess-2019-353/nhess-2019-353-AC1-supplement.pdf

––––––––––––––––––––––––––

[Figure]

**Supplement:**

GENERAL COMMENTS In this paper, the authors analyse sea level and skew surge extremes (values greater than the 5-yr return level are considered extremes) from 30 tide gauges around the coast of New Zealand. The objective is to characterise the frequency and magnitude of these extreme events andalsoestimatethecontributionofeachsealevelcomponent(tide,surgeandMSLA) to the sea level extremes. Sea level rise is not taken into account in this study. In my opinion, the manuscript is very well written, well structured, clear and easy to understand. However, I have one main scientific concern and a few smaller issues that are explained below.

SPECIFIC COMMENTS I have one main concern regarding the extremes' methodology: The authors say they use a GPD+POT model to obtain the return levels, but they define POT as the 5 largest events per year (line 99), which in my opinion is the r-largest method and not POT. The POT method should keep a constant threshold through time and in this case it varies each year. I think that extremes selected with the r-largest method should be fitted to a GEV instead to a GPD, so I am not sure if the return levels obtained here are correct.

*Response: We did use the POT method with a fixed height threshold for each site but had not accurately explained it in the text, which made it appear as if we used an r-largest method when in fact we used POT. We have amended the text as shown below, to accurately describe what we did. For further confirmation, in* Table 1 *below we have also included the number of maxima exceeding the (1.69 m) threshold at Auckland, wherein the number of exceedances of the threshold is seen to vary year to year.*

130

135

140

To identify the 1 in 5-year return levels the extremes of the distributions were modelled using a generalized Pareto distribution (GPD) fitted to peaks-over-threshold (POT) data (Coles 2001).  was used to determine the 1 in 5-year return level thresholds for identifying extreme storm-tide and skew-surge event, because the GPD model is fitted directly to the independent storm-tide and skew-surge maxima (e.g., Figure S1) associated with storms that we analyse later. POT consisted of sea-level maxima  with the maxima being separated by a minimum of 3 days, since separate meteorological systems generally pass over NZ with 4–7 days of each other. A test using a 2-day threshold for both storm-tide and skew-surge showed that the identification of extreme events was insensitive to the time threshold. The POT height threshold was selected to give an average of approximately 5 high-water maxima per year over the duration of the measurement record, which is equivalent to about the 99.8th percentile of the hourly data (the number of maxima exceeding the height threshold varies year-to-year).

lines 115-116: It is difficult to extract this information from table S7, maybe those TGs longer than 50 yr could be highlighted or the table could be order by the TGs length instead of the site number?

*Response: Rather than refer to Table S7, we have added a new supplementary Figure S1 (other supplementary figures sequentially renumbered) which demonstrates how the SSJPM and GPD models are similar for extreme sea levels ≥ 5-year return period but the SSJPM generally better matches the larger outlying maxima in the longer records.*

Table S7: It is not clear to me what the "model percentile" means in table S7. Are 2.5% and the 97.5% the confidence intervals? maybe the table could be simplified using the GPD or SSJPM values +- the confidence intervals?.

*Response: We have added a description to the caption to make it clear that 50% = median of the fitted distribution, 2.5% = lower 95th percent confidence interval, 97.5% = upper 95th percent confidence interval. Simplifying the table using a ± confidence interval is not possible because the confidence limits are not symmetrical about the median. We have simplified the Table to remove the GPD results, following review comments from Franck Mazas.*

Why is it that this information does not appear in table S8 for the skew surge?.

*Response: the SSJPM method cannot be applied to skew-surge, which is a component of extreme sea level. But we have now included the 95% confidence intervals for the skew-surge in Table S8. Thanks.*

Another main concern I have is that the return level estimations should not exceed 4 times the length of the observations (Pugh & Woodworth, 2014), since the longest TG is 120 yr, the 1000 return level is not reliable for any location. Maybe the reliable return levels, at each location, could be highlighted. Also, if sea level rise is not included, maybe there is no point in obtaining such large return periods.

*Response: This is a good point and so we have provided some guidance to temper the use of the return level estimates. Using GPD the reliable return levels are about 4 times the record length, but this can be about 10 times the record length for joint-probability methods (see Haigh et al. 2010). Therefore, immediately preceding Table S7, we have added a paragraph to guide the use of Table S7, which says:*

> *"Table S7 presents extreme storm-tide return period height estimates out to 1000-year return period. Whereas direct GPD estimates are generally only reliable for return periods up to 4 times the record length (Haigh et al. 2010; Pugh & Woodworth, 2014)†, joint-probability (e.g., SSJPM) estimates can be statistically reliable to about 10 times the record length (Haigh et al. 2010). The record lengths are included in the table to guide the use of extreme storm-tide return period height estimates. For example, the 6-year record length at Whangaroa up to 50-year return period using the SSJPM."*

*We note that return periods are surrogate for probability of occurrence, e.g., 1000-year return period is equivalent to 0.001 annual exceedance probability. Return periods (aka probabilities of occurrence) should not be confused with time periods over which sea-level rise might act. There are several papers showing that show (even quite small) sea-level rise will dramatically change the probability of exceedance of sea-levels reaching a fixed height relative to a land-based datum, but it is still relevant to provide low exceedance probability sea-level height estimates at present-day mean sea level— quantifying how these exceedance rates could change (e.g., Hunter 2010, Sweet & Park 2014, Stephens et al. 2018) is an extra step beyond the present research paper.*

lines 134-135: I am not sure if I understand: Some of the 155 SL extremes are not independent, so after keeping only those separated more than 3 days, it results in 85 independent events, is that right? (same for the Skew surges?). Also, which methodology (GPD or SSJM) is used for obtaining the return periods in table S2 and S3?.

*Response: We have rearranged the text to clarify, as below:*

We compared the dates and return periods for both extreme  storm-tide and extreme skew-surge events, and ascertained how many skew-surge events also resulted in extreme  storm-tides.

180     Extreme storm-tides  were generated by 85 distinct storm events (Figure 2a, Table S2). These 85 storm events contributed to 155 independent storm-tide maxima that  reached or exceeded the 1 in 5-year return level  at any of the 30  sea-level-gauge sites (Figure 1)—in other words the storm events sometimes caused extreme storm-tides to occur at more than one sea-level gauge site. In total, 191 skew-surges reached or exceeded the 1 in 5-year return period across the 30 sites, generated by 135 distinct

185    storm events (Figure 2b, Table S3). This difference in event numbers is consistent with NZ's coastal context of meso- to micro-tidal ranges and moderate storm surges, which are usually several decimetres (not metres) high, meaning tides are a key determinant (or precursor) for coastal flooding rather than solely storm surges.

*Response: The captions of Tables S2 and S3 now describe which method was used. Thanks.*

line 163: from figure S2 I am not able to infer those values. For example, the ratio for station 18 should be aprox. 1.2m/2m, right?, but figure S2c reads ratio equal to 1.

*Response: In plot c we had mistakenly plotted maximum skew-surge / MHWS-7. This has now been corrected. Thanks.*

lines 169-170: Are those equations used somewhere else? what info can we infer? maybe they are not needed?

*Response: The equations are not used elsewhere in the paper but might be of relevance to a user conducting local studies within NZ, so we prefer to leave them as is.*

figures S3b and S4a shouldn't be identical?

*Response: Figure S3b and S4a (S4b and S5a in revised supplementary information) shouldn't be identical and are not identical. S3b shows high tide at Auckland and S4a shows mean storm-tide elevation throughout NZ.*

figures 7a and S7a shouldn't be identical?

*Response: We deliberately reproduced Figure 7a within Figure S7a for comparison with Figure 7b, to enable the reader to make a quick visualisation of the impact of MSLA on the seasonal distribution of extremes, without having to flick back to the paper .*

lines 252-254: I am not able to extract this information from figure 8a

*Response: The information can be inferred from Figure 8a—there are no events (blue dots) falling within 4–10 days on the y-axis? Haigh et al. 2016 conveyed similar information using the same type of plot (Figure 6, Haigh et al. 2016).*

line 269: maybe this event could be highlighted in figure 3a

*Response: We have altered the text as per below to point to the event in Figure 3a. We have also updated the numbers relative to those originally transferred from Stephens et al. (2014) to report skew-surge instead of storm-surge and have updated the value for MSLA (Stephens et al. 2014 used a wavelet filter rather than a 30-day running average).*

330  causing millions of dollars of flood-related damage. At 0.4 52 m the  skew- surge was  relatively large for NZ  and coincided with a predicted "red alert" (high perigean-spring) tide, and a background MSLA of nearly 0.1 05 m , noting that ongoing SLR increasingly adds to the impact or frequency of flooding (Stephens et al. 2018). SLR of about 0.12 m between 1936 and 2011 means that after detrending for sea-level rise the 2011 storm-tide is actually the 2nd-largest storm-tide with the 26 March 1936 storm-tide being largest (Figure 3a).

335

TECHNICAL CORRECTIONS

Figure 4: x-axis is MHWS or MHWS-7?. y-axis is extreme sea level? I think storm-tide is not defined in the paper.

*Response: We have re-labelled the axes. Storm-tide is now defined in the first paragraph of the Data and Methods section.*

line 219-220: September-December?

*Response: Correct, thanks—text amended to September–December.*

line 221: April-July?

*Response: March–July are consecutive months that have higher monthly occurrences of ≥5-year ARI skew-surges than any other month*

line 228: April-July?

*Response: March–June is correct based on the peak of the mean annual cycle (not shown).*

**Franck Mazas (Referee)**

Overview The present paper addresses the spatial and temporal patterns of extreme sea levels and skew surges in New Zealand, which is interesting, not so much for providing local extreme values but rather for understanding the conditions in which such events occur. Accounting for MSLA in particular provides valuable information, and finding that the seasonal patterns of extreme sea levels follow the seasonal pattern of MSLA rather than that of astronomical tide is very interesting.

The paper is well written and structured, and the authors are well known in the field of research. Although it could be directly accepted, I have identified several ways of improving it.

*Response: Thanks for the feedback Franck. We have acknowledged your helpful review in the paper.*

Specific comments

First, the reader would probably welcome, after the introduction section, a short description of tides and surge climate in New Zealand. Is the tide semi-diurnal, with diurnal inequality, mixed...? What is the typical tidal range, let us say along both the east and west coast? What are the typical surge values? Is there a wide or a narrow continental shelf? Some of these information are provided here and there in the text, but having a good (though concise) overview before beginning the data / methodology section would help.

*Response: Thanks for the feedback. We have added a section "New Zealand storm-tide characteristics".*

As regards the extrapolation technique, it looks like the sampling method is the Rlargest method, not the POT one. So in this case there is no Poisson assumption, and this is generally a GEV distribution that is fitted. But maybe is it 5 values per year in average? However, I am more bothered by the choice of considering equally a direct and in indirect approach for extreme sea levels (even more when one of the co-authors has shown the differences between both approaches!). I do not really see what the results of the direct POT (or R-largest) extrapolations provide to the paper, apart from confusion.

*Response: We did use the POT method with a fixed height threshold for each site but had not accurately explained it in the text, which made it appear as if we used an r-largest method when in fact we used POT. We have amended the text as shown below, to accurately describe what we did. To further illustrate, in Table 1 below we have also included the number of maxima exceeding the (1.69 m) threshold at Auckland, wherein the number of exceedances of the threshold is seen to vary year to year.*

A generalized Pareto distribution (GPD) fitted to peaks-over-threshold (POT) data (Coles 2001) was used to determine the 1 in 5-year return level thresholds for identifying extreme storm-tide and skew-surge event, because the GPD model is fitted directly to the independent storm-tide and skew-surge maxima (e.g., Figure S1) associated with storms that we analyse later. POT consisted of sea-level maxima  with the maxima being separated by a minimum of 3 days, since separate meteorological systems generally pass over NZ with 4–7 days of each other. A test using a 2-day threshold for both storm-tide and skew-surge showed that the identification of extreme events was insensitive to the time threshold. The POT height threshold was selected to give an average of approximately 5 high-water maxima per year over the duration of the measurement record, which is equivalent to about the 99.8th percentile of the hourly data (the number of maxima exceeding the height threshold varies year-to-year).

*Considering the feedback we agree that presenting the results of two distributions is unnecessary and we prefer the SSJPM for prediction of extreme sea-level frequency and magnitude so we have removed the GPD results from Table S7. To justify this we have added a new Figure S1 to compare the results of GPD and SSJPM at the longest sea-level records, which shows that the SSJPM generally better matches the larger outlying maxima in the longer records. But it is still necessary to use a direct maxima approach (GPD) to calculate the frequency and magnitude of skew-surge. Also, we still use the GPD/POT to identify recorded extreme storm-tide "events" that exceed the 1/5-year threshold as described in the paper excerpt below. The reason is that the GPD is directly fitted to POT and the sampling of POT is directly consistent with the identification of the events that we later analysed. Ultimately is doesn't matter much whether the GPD or SSJPM was used to calculate the 1/5-year threshold above which sea-level events are considered extreme or not for later analysis of spatial and temporal patterns, however, doing this again using a SSJPM-based threshold would cause a slightly different set of events to be selected and require a comprehensive rework of the datasets but would not affect the key results or conclusions. We have explained this in the text.*

al. 2011). In summary, the GPD/POT model was used to determine extreme skew-surge frequency and magnitude and to identify extreme storm-tides and skew-surges with ≥ 1 in 5-year return levels for further analysis. We used  the SSJPM to determine the return period of extreme storm-tides (e.g., Figure 2a, Tables S2, S4 & S7) when the GPD and SSJPM methods both indicated a ≥10-year return period for a particular storm-tide, since we prefer the SSJPM method but there can be a mismatch between the GPD and SSJPM at the 5-year return level used for storm-tide maxima selection (Figure S1). Using a SSJPM-based storm-tide threshold would select a slightly different set of storm-tides for analysis but would not affect the key results or conclusions.

I have seen at several occurrences the (widely spread) confusion between level, a vertical position that is always referenced to a datum such as LVD, CD or MSL, and height, a vertical distance, in m, independent of the datum (a height being a difference between two levels). In particular, comparisons between a surge (which is an extra height) and a sea or tidal level is, strictly speaking, physically meaningless. Therefore, figures and tables referring to "sea-level (m)" cannot be understood. For the purpose of this paper, it seems that the relevant physical quantities are the tidal amplitude, or semi-range (difference between tidal level and MSL), skew surge (unchanged), MSLA (unchanged)and what could be referred to as "sea-level height"(or, preferably, abetter name),namely the difference between sea level and MSL. Therefore, "sea-level height" would be the sum of tidal amplitude and skew surge (which includes MSLA), and all quantities could be compared. This could be explained in the first sections, before the results.

*Response: Thanks. We have added a paragraph at the start of Section 2 to define "storm-tide" as the variable of interest. We have amended the text, figures and tables to use this unambiguous term.*

Last, it is explained that "red-alert" forecasts are emitted to coastal and hazards managers, because extreme sea levels tend to occur in conjunction with high tides, as shown in this study. But considering the other findings of the study and the relationships with the weather types, the reader would think that it should be rather easy and straightforward to improve this warning system by combining the occurrence of high tides with the weather types prone to surges. This would be an immediate benefit of the study, and it seems strange that this possibility is not mentioned.

*Response: We have added text around the red-alert tide calendar discussion:*

315 approximately 25% of the average tidal range (Stephens et al. 2014). Coastal or hazard managers are advised to keep a close watch on the weather for lower barometric pressure and adverse winds during the red-alert tide days, as even a minor storm or swell event could lead to inundation of low-lying areas, especially if accompanied by waves (Bell 2010). In practice, weather forecasts alert managers to high surges and storm-tides with <10-days notice, but the red-alert tide calendars enable resource planning (e.g. staff leave days) months in advance of potential high-storm-tide days and are valued coastal climate service in

320 NZ. For example, the highest storm-tide in Auckland (NZ) on record since 1900 occurred on 23 January 2011, closing major

- l. 62: "... are well sampled by tide gauges": if no seiching occurs, or, if it does occur, if the sampling rate is fine enough to characterize these LF waves

*Response: Thanks, we have included your suggestion.*

- l. 79/80: how many components for the harmonic analysis?

*Response: 67—now described in the text.*

- l. 81: "annual and semi-annnual tides" -> "... components"?

*Response: "tidal constituents" now in text*

- l. 82: "...most of the seasonal signal is actually driven by non-astronomical effects": well, at least not directly, I believe this is a longstanding debate... But this is not very important.

- l. 93: this is not the best definition of return period, it would be better to speak of a 1/5 annual probability of exceedance, while highlighting the cumulative effect, year after year, and the probability of encounter in a given period of several years.

*Response: The best definition of return period depends on the context. In our work for clients in NZ we favour AEP over return period, because the concept of return period can be confusing for the public when for example several long-return period events occur in close succession. Because return period is measured in years it is also easily confused with planning timeframes for sea-level rise. One advantage of return period though, is that the magnitude of the event scales with the return period—both get bigger at the same time, whereas AEP reduces with event magnitude. It was for this reason that we deliberately framed the paper in terms of return period because in our opinion it allows clear presentation of the arguments within the context of the paper, and is also directly comparable with Haigh et al. (2016). This paper focuses on historical extreme events and does not address future frequency change after a period of SLR so confusion with planning timeframes is a non-issue. We have addressed the cumulative effect, year after year, and the probability of encounter in a given period of several years in another paper (Stephens et al. 2018).*

- l. 113-116: simple statistics exist for assessing the tide-surge dependence, see in particular Dixon and Tawn (1994)

*Response: Tests of independence (Dixon & Tawn 1994; Haigh et al. 2010) that we have done in the past (not shown in the paper) show that all sea-level records in NZ exhibit some tide and storm surge dependence—the null hypothesis (no interaction) was not satisfied for any sites, even those on the open coast, although these sites have the lowest $\chi^2$ value. However, the dependence referred to is between hourly tide and hourly storm surge and NOT skew-surge. Batstone et al. (2013) and Williams et al. (2016) use the SSJPM because skew-surge is largely independent of tide even while tide and storm surge is dependent.*

-Figure2, legend: if I understand, well, these are not the "85 extreme sea-level events" or "135 skew-surge events", but rather the 85 (135) storms generating the extreme sea levels (skew surges)

*Response: We have added a definition of the word "event" to the text and checked and amended use of the word event throughout. We have also amended the Figure 2 caption for clarity. Thanks*

> 125    tide or on a neap tide. In this paper an "event" is defined as a meteorological storm that caused an extreme storm-tide or skew-surge at least one tide gauge but often at several tide gauges—so a single "event" can have multiple observed extremes.

- l. 162/163: "ratio of maximum observed skew surge to maximum observed sea level": only relevant if considering the elevation relative to MSL, see comment above

*Response: This is OK because the ratios are relative to MSL—we are now using the term storm-tide to make this clear.*

-Figure3: they-axis cannot be "sealevel (m)", but something like "surge/water height above MSL", see comment above

*Response: Now that the paper has been reframed to use the term storm-tide for the extreme sea-level events, then "Sea-level (m)" is now an appropriate axis label because the plot shows various components of sea level?*

- l. 169/170: same comment, is it a comparison of levels in m LVD, or of water height above MSL?

*Response: Also corrected by use of the term storm-tide.*

- l. 260: "despite the fact that the UK has larger tides": yes, but maybe the ratio surge / tide is similar?

*Response: Now addressed in the text.*

- l. 264/266: interesting information, but it is mentioned too late in the paper (see comment about a description of tide and surge climate in NZ)

*Response: This information is now mentioned early in the paper in Section 2.*

- Section 4.3, §2 and 3: the logical link between the two paragraphs is not clear to me

*Response: I'm not sure how to respond. Need they be linked? There are 5 paragraphs in Section 4.3 (now 5.3) that discuss distinct aspects of the role that surge plays.*

- l. 325/326 (hourly surges), in relation with l. 336/338 (numerical models): indeed, if the total signal of surge can be accurately distinguished (sometimes very difficult with gauge measurements, hence the skew surge approach, but straightforward through numerical modelling), then much more

information can be used for probabilistic extrapolation. Mazas et al. (2014) provide a POT-JPM model that works for the total signal of skew surge and hourly (or 10 min) residual as well.

*Response: Stephens et al. (2018) adopted a similar method and cited Mazas et al. (2014), but we have used GPD and SSJPM in this paper.*

*Table 1.* Number of peaks over threshold ($N_{POT}$) per calendar year at Auckland

| Year | $N_{POT}$ | Year | $N_{POT}$ | Year | $N_{POT}$ | Year | $N_{POT}$ | Year | $N_{POT}$ |
|------|-----------|------|-----------|------|-----------|------|-----------|------|-----------|
|      |           | 1926 | 4  | 1951 | 3  | 1976 | 5 | 2001 | 8 |
|      |           | 1927 | 3  | 1952 | 4  | 1977 | 4 | 2002 | 2 |
| 1903 | 1  | 1928 | 7  | 1953 | 3  | 1978 | 5 | 2003 | 4 |
| 1904 | 7  | 1929 | 4  | 1954 | 6  | 1979 | 7 | 2004 | 0 |
| 1905 | 8  | 1930 | 3  | 1955 | 6  | 1980 | 3 | 2005 | 2 |
| 1906 | 2  | 1931 | 1  | 1956 | 12 | 1981 | 5 | 2006 | 5 |
| 1907 | 6  | 1932 | 1  | 1957 | 10 | 1982 | 1 | 2007 | 1 |
| 1908 | 4  | 1933 | 2  | 1958 | 4  | 1983 | 1 | 2008 | 4 |
| 1909 | 10 | 1934 | 5  | 1959 | 7  | 1984 | 5 | 2009 | 5 |
| 1910 | 7  | 1935 | 7  | 1960 | 4  | 1985 | 2 | 2010 | 4 |
| 1911 | 8  | 1936 | 5  | 1961 | 7  | 1986 | 0 | 2011 | 5 |
| 1912 | 7  | 1937 | 7  | 1962 | 7  | 1987 | 2 | 2012 | 7 |
| 1913 | 5  | 1938 | 11 | 1963 | 4  | 1988 | 1 | 2013 | 7 |
| 1914 | 2  | 1939 | 0  | 1964 | 4  | 1989 | 5 | 2014 | 9 |
| 1915 | 1  | 1940 | 7  | 1965 | 3  | 1990 | 1 | 2015 | 8 |
| 1916 | 5  | 1941 | 5  | 1966 | 4  | 1991 | 1 | 2016 | 9 |
| 1917 | 7  | 1942 | 0  | 1967 | 2  | 1992 | 3 | 2017 | 5 |
| 1918 | 7  | 1943 | 7  | 1968 | 2  | 1993 | 3 | 2018 | 5 |
| 1919 | 8  | 1944 | 4  | 1969 | 0  | 1994 | 1 |      |   |
| 1920 | 7  | 1945 | 5  | 1970 | 3  | 1995 | 4 |      |   |
| 1921 | 3  | 1946 | 0  | 1971 | 6  | 1996 | 4 |      |   |
| 1922 | 8  | 1947 | 5  | 1972 | 9  | 1997 | 7 |      |   |
| 1923 | 10 | 1948 | 6  | 1973 | 2  | 1998 | 6 |      |   |
| 1924 | 10 | 1949 | 4  | 1974 | 5  | 1999 | 4 |      |   |
| 1925 | 5  | 1950 | 5  | 1975 | 3  | 2000 | 3 |      |   |

---

## Author Comment (AC2) · 28 Jan 2020

The comment was uploaded in the form of a supplement: https://www.nat-hazards-earth-syst-sci-discuss.net/nhess-2019-353/nhess-2019-353-AC2-supplement.pdf